# Research on differential game strategy of debt restructuring supported by government

**Danyu Zhao[1], Li Song[1]\*, Liangliang Han[2]**

**1** School of Management, Shenyang University of Technology, Shenyang, 110870, Liaoning, PR China,
**2** Business School Liaoning University, Shenyang, 110136, Liaoning, PR China

\* songli@sut.edu.cn

## Abstract

This paper studies the debt restructuring equilibrium decision problem composed of creditors and debt enterprises with the participation of the government and asset management companies. With the differential game, the dynamic optimization models of debt restructuring under three situations: centralized decision-making, decentralized decision-making, and Stackelberg game after introducing cost-sharing contract are constructed, respectively. The optimal equilibrium strategy of debt restructuring, the optimal trajectory of debt restructuring synergy, and the optimal profit under three decision-making situations are investigated and compared. It is found that the synergy effect and total profit of debt restructuring are the highest under centralized decision-making, and the Stackelberg game is superior to decentralized decision-making, which shows that the cost-sharing contract can achieve the coordination of overall interests, improve the debt restructuring environment, and promote the debt restructuring process. Finally, the sensitivity analysis of relevant parameters is carried out through an example, which verifies the effectiveness of the conclusion and provides the scientific basis for the government and asset management companies to participate in debt restructuring successfully.

## Introduction

The international financial crisis broke out, and the debt dilemma of enterprises became more and more serious. The financial ecological environment continued to deteriorate, and listed enterprises enter the peak of debt restructuring. The existence of a large number of non-performing assets in enterprises leads to the decrease in liquidity and the aggravation of asset loss. It can not bring the expected income to the enterprise, which leads to the decline of the competitiveness of enterprise assets. The market activities and economic competition of enterprises are becoming increasingly fierce, and a good economic level is very important for improving social welfare, which also shows the importance of properly handling enterprise debts [1–4]. Since 2013, the international economic environment has changed with inflection point significance. However, the non-performing assets of Chinese enterprises rise instead of falling, and the enterprises are in a state of stagnation and lack liquidity, so it is difficult to bring expected returns to the enterprises, and the loss of assets is extremely serious. The above problems have led to the decline of the overall asset quality of China enterprises, and further stifled the

**Funding:** This research was funded by National Social Science Foundation of China, grant number

21BGL110. Liangliang Han: The funders had no role in study design, data collection and analysis, decision to publish, or preparation of the manuscript.

**Competing interests:** The authors have declared that no competing interests exist.

competitiveness of Chinese enterprises in the international market, causing a series of vicious cycles of enterprises debt difficulties [5–7].

Debt restructuring is an effective means to solve the financial difficulties of enterprises. Many enterprises have brought new vitality to their development through debt restructuring. As an important economic activity, debt restructuring is one of the important means for enterprises to reduce financial risks and integrate social resources [8,9]. In recent years, the ratio of enterprises debt restructuring has shown a clear upward trend, which has aroused widespread concern in the economic community [10]. Nowadays, countries worldwide have completed a large number of debt restructuring based on the debt situation of non-performing assets of their enterprises. Sweden, Finland, and other countries have turned their enterprises from losses to profits through debt restructuring. In Sweden, in the early 1990s, the Swedish government took supporting enterprises as the starting point to help banks carry out debt restructuring, set up asset management companies, and dispose of non-performing assets, helping the Swedish banking industry gradually out of the trough. The Finnish government rescued banks and enterprises through debt restructuring, supported the development of small and medium-sized enterprises through the stripping and selling non-performing assets of banks, and finally successfully got out of the debt dilemma of non-performing assets. In China, there are also many successful cases of enterprises successfully disposing of non-performing assets through debt restructuring [8–10]. In recent years, many important enterprises related to the national economy and people's livelihood have been reborn through debt restructuring, such as Sanpower Group and Luckin coffee. These enterprises improve their business development and financial situation through debt restructuring. In view of this, debt restructuring should be the best choice for enterprises when they are deeply mired in non-performing assets and debts. Compared with bankruptcy reorganization, reconciliation and bankruptcy liquidation, debt restructuring can better protect the interests of both creditors and debtors, make both sides achieve win-win and maximize their benefits. Therefore, this has also become a direct reason for Chinese enterprises to choose to implement debt restructuring activities when they are in financial difficulties.

When the enterprise has a debt crisis, from the perspective of creditors, if the debt enterprises choose bankruptcy liquidation, and the creditors require its bankruptcy liquidation to repay the debt, it will generate a lot of sunk costs and lose the interests of creditors. The creditor may agree to the debtor to repay the debt in a way lower than the original debt conditions by agreement, and creditors can recover their claims to a certain extent without causing greater losses. On the other hand, debt enterprises will have the opportunity to continue to operate, restore their hematopoietic capacity, and creditors will also get more benefits. For the debtors, debt restructuring can relieve the pressure of debt repayment, improve their operating function, increase the total profits of the current period, effectively solve the financial difficulties and debt crises of enterprises, maintain social stability, and become possible to reverse losses [5]. Based on this, the government should try its best to help enterprises reduce the debt burden, avoid enterprises using new loans to repay previous loans, falling into a vicious cycle of lending, and formulate effective debt management policies to improve economic productivity [11].

In terms of policy, the Chinese government strongly supports debt restructuring of enterprises in debt distress, increases tax rebate policy, provides tax rebate and tax reduction policy, solves difficulties for enterprises in debt distress, expands the scale of government financing guarantee business, and reduces financing guarantee rate. At the same time, in order to help enterprises tide over difficulties, the government has also increased its procurement efforts from small and medium-sized enterprises to help enterprises increase business volume and alleviate the debt crisis [8]. In terms of finance, accelerate the issuance and use of local

government special bonds, and the financial department will revitalize the stock of local government special debt limits according to law, and give better play to the role of special bond funds; Delaying administrative fees, supporting the use of policy development financial instruments, promoting the orderly landing of important projects, and helping enterprises to reduce their burdens. Through debt restructuring, the distressed enterprises with room for rescue will be restructured to relieve the debt pressure of distressed enterprises, retain their development potential and inject operational vitality as much as possible.

At present, some scholars have studied the impact of debt restructuring decisions. Suet et al. [1] believe that reaching a debt restructuring agreement is a decision-making process. The creditor will decide the way of debt renegotiation according to the situation, the government has a certain influence on the bank's personnel appointment and removal and major decisions, the debtor's decision will also affect the debt restructuring process. Mo et al. [12] find that as the strength of political connection increases from weak to strong, the positive impact of debt restructuring on investment turns to a negative impact. Bergström et al. [13] indicate that restricting the priority of secured debt may stimulate debt restructuring, and there is a negative correlation between the security of financial institution lenders and the possibility of debt restructuring. Relevant scholars conduct research on debt pricing [5–9], such as Kim et al. [6] constructed a liquidity default model. Anderson et al. [8] (1996) constructed a strategic debt payment model. Leland et al. [9] constructed default models caused by economic distress. Macas et al. [14] found that the more companies that use debt in price agreements, the more unstable the price agreements are. Acharya et al. [15] (2006) tried to combine strategic debt payment with equity refinancing and introduce equity refinancing to solve the problem of insufficient liquidity, but were limited to establishing a static model.

Based on the above research results, the existing research mainly focuses on the influencing factors and debt pricing of debt restructuring. However, the research results on the scientific decision-making of each participant in the debt restructuring process and the interaction of various strategic influencing factors are rare. The application of mathematical modeling method in multi-agent decision-making of debt restructuring is still in its infancy. This paper studies debt restructuring in a new way, fills the gap in the multi-agent decision-making theory of debt restructuring, tries to improve the debt restructuring system, interprets the multi-agent decision-making problem of debt restructuring from a new perspective, opens up a new path for the multi-agent decision-making research of debt restructuring, optimizes the existing research on debt restructuring, and enriches relevant literature. The debt restructuring process is a complex and dynamic process resulting from the familiar game of all participants. Due to information asymmetry, it is difficult for each participant to achieve a specific equilibrium in a decision, and the interaction between participants is needed to achieve the final equilibrium. The differential game just makes up for the deficiency of the traditional game method. By extending the game theory to continuous time through differential games, each participant can change their strategy in an infinitely short time period, so as to solve a dynamic model of multi-party cooperation or competition in continuous time. Therefore, this paper uses the method of the differential game to construct the debt restructuring game model between creditors and debt enterprises with the participation of the government and asset management companies to help all parties find the Pareto optimal equilibrium decision of system profits.

The contributions of this paper are as follows: (1) This paper studies the balanced decision-making of debt restructuring with multi-agent participation from a dynamic perspective, providing correct theoretical guidance for promoting the success of debt restructuring. (2) From the perspective of game theory, using the method of differential game, this paper constructs the differential game model of debt restructuring between creditors and debt enterprises with the participation of the government and asset management companies. Through the analysis

of centralized decision-making, decentralized decision-making and decision-making after the introduction cost-sharing contract, this paper investigates the optimal effort level of participants, the synergistic effect of debt restructuring, the interests of all participants and the overall income of debt restructuring system in three situations, and finds the optimal decision-making mode and mechanism of debt restructuring. (3) This paper fills the blank of debt restructuring in multi-agent decision-making theory, tries to improve the debt restructuring system, interprets the multi-agent decision-making problem of debt restructuring from a new perspective, opens up a new path for the research of debt restructuring multi-agent decision-making, optimizes the existing research of debt restructuring, enriches the theoretical system of debt restructuring, and has an excellent expansion effect on the relevant theories of debt restructuring multi-agent decision-making. (4) The debt restructuring process is inseparable from the participation of the government and asset management companies. This paper studies the problem of government subsidy to debt restructuring parties and the debt preference given by asset management companies. On this basis, it discusses the income distribution mechanism between the two sides of debt restructuring, analyzes the efforts of debt enterprises and the influence of the strategies chosen by creditors on the performance of debt restructuring, and provides the theoretical basis for the government to formulate subsidy policies.

## Model construction and solution

### Problem description

This paper takes the debt restructuring system as the research object, and the fundamental goal is to maximize the interests of both parties. In the debt restructuring system, the game relationship between the main bodies is objective, and in practice, the game between the main bodies has both cooperation and competition. In the process of debt restructuring, many subjects are often involved, such as creditors, debt enterprises, asset management companies and the government. Among them, the ultimate goal of the government is to help enterprises avoid bankruptcy liquidation to maintain social and economic stability, the ultimate goal of banks is to reduce bad debt losses, the ultimate goal of enterprises is to reduce debts, and the ultimate goal of asset management companies is to maintain social stability and improve their own interests.

In this paper, only creditors and debt enterprises are taken as the research subjects to conduct modeling and analysis to study the impact of the strategic choices of creditors and debt enterprises on the debt restructuring system. They are mutually restricted and cooperative, and the efforts of creditors to cooperate with restructuring directly affect the success rate of debt restructuring, while the efforts of debt enterprises directly affect the debt restructuring process. This paper chooses the creditor as the dominant party of the differential game, and forms Stackelberg differential game with debt companies. Under the circumstances of centralized decision-making, decentralized decision-making and Stackelberg game, this paper studies the effort cost of both debt enterprises and creditors, the optimal trajectory of debt restructuring synergy and the profits of all parties and the whole, the success of debt restructuring depends on the joint efforts of all participants. In order to encourage all restructuring parties to actively cooperate with debt restructuring, the government will give policy incentives, and asset management companies will give certain financial support. Therefore, this paper takes government subsidies and the concessions given by asset management companies as exogenous variables to study their influence on debt restructuring decisions.

### Model assumptions

Hypothesis 1: Consider the debt restructuring system composed of a single creditor ($M$) and a single debt enterprise ($N$). Assuming that the participants are entirely rational, all of them aim

at maximizing their own benefits, and have complete information, the effort level of creditors participating in debt restructuring is $E_M(t)$, and the effort level of debt enterprises is $E_N(t)$, which are the decision variables of creditors and debt enterprises respectively. Assume $\mu_M$ and $\mu_N$ respectively represent the effort cost coefficients of creditors and debt enterprises, $C_M(t)$ and $C_N(t)$ respectively represent the effort cost of creditors and indebted enterprises, which are convex functions of their respective efforts. In this paper, referring to the functional expression of reference [10], we set it as a quadratic function, that is, the functional expressions of both parties' effort costs $C_M(t)$ and $C_N(t)$ are as follows:

$$C_{M(t)} = \frac{\mu_M}{2} E_M^2(t); \ C_{N(t)} = \frac{\mu_N}{2} E_N^2(t) \tag{1}$$

Hypothesis 2: The level of efforts of creditors and debt companies has a positive impact on the synergies of debt restructuring, references [10–12], it is assumed that the synergistic effect decays with time, that is, there is a natural decay rate $\alpha$ and $\beta$ are the influence coefficients of the debt restructuring efforts of creditors and debtor enterprises on the synergistic effect of debt restructuring, among $\alpha > 0, \beta > 0$; $\gamma$ is the natural decay rate of the synergistic effect, where $\gamma > 0$; $K(t)$ represents the level of synergy between creditors and debt enterprises, where $K(0) = K_0 \geq 0$ is the synergy of initial debt restructuring, the differential equation for the synergistic effect of debt restructuring is:

$$\dot{K}(t) = \alpha E_M(t) + \beta E_N(t) - \gamma K(t) \tag{2}$$

Hypothesis 3: References [14–16], at time $t$, the total revenue $\psi(t)$ of the debt restructuring collaborative system of creditors and debt companies can be expressed as:

$$\psi(K(t), t) = \omega K(t) + \psi_0 \tag{3}$$

Among them: $\omega$ represents the degree of influence of the synergistic effect of debt restructuring on the total income.

Hypothesis 4: This paper focuses on the efforts level of creditors and debt enterprises to participate in debt restructuring. In order to simplify the model, other factors affecting the return of debt restructuring are not considered, and both parties make decisions based on complete interest.

Hypothesis 5: The total income of the debt enterprises and the creditors participating in the debt restructuring is only distributed between the creditor and the debt enterprise, and the income ratio of the creditor $M$ is $\theta$, and the income ratio of the debt enterprise $N$ is $1-\theta$, and the income distribution ratio is determined by the two parties in advance.

Hypothesis 6: Government subsidies are an important component of fiscal expenditure, and debt restructuring is an important way to avoid bankruptcy liquidation of enterprises. Government subsidies can reduce the cost of hard work for enterprises. $\varphi_M$ and $\varphi_N$ are used to represent the government's subsidy rate for $M$ and $N$.

Hypothesis 7: As an important participant in the process of debt restructuring, the asset management company negotiates with the debtor company about the specific process of debt restructuring after taking over the debt of the creditor, and the enterprise can obtain certain financial support. Therefore, the participation of the asset management company can effectively reduce the effort cost of the enterprise and the creditor. Among them, the proportion of reducing the debt restructuring cost of creditors is $\tau$, and the proportion of reducing the restructuring cost of debt companies is $\sigma$.

Hypothesis 8: The hypothesis of the references in this paper [17,18], in the infinite time interval, both parties have the same discount factor $\rho$ at any time.

To sum up, the long-term profits of the creditors, debt enterprises and the debt restructuring system can be obtained as follows:

$$P_M(x) = \int_0^\infty e^{-\rho t}\left[\theta\psi(t) - \frac{\mu_M}{2}E_M^2(t) + \varphi_M\frac{\mu_M}{2}E_M^2(t) + \tau\frac{\mu_M}{2}E_M^2(t)\right]dt \tag{4}$$

$$P_N(x) = \int_0^\infty e^{-\rho t}\left[(1-\theta)\psi(t) - \frac{\mu_N}{2}E_N^2(t) + \varphi_N\frac{\mu_N}{2}E_N^2(t) + \sigma E_N^2(t)\right]dt \tag{5}$$

$$P_T(x) = \int_0^\infty e^{-\rho t}\left[\psi(t) - \frac{\mu_N}{2}E_M^2(t)(1-\varphi_M-\tau) - \frac{\mu_M}{2}E_N^2(t)(1-\varphi_N-\sigma)\right]dt \tag{6}$$

For writing convenience, $t$ is omitted below.

## Centralized decision-making

Centralized decision-making (Indicated by the superscript $A$) emphasizes the profit maximization of decision-makers as a whole. That is, creditors and debt enterprises negotiate to determine the level of debt restructuring efforts, with the goal of maximizing the overall profits of both parties, so as to improve the competitiveness of debt enterprises. At this point, the decision objectives are:

$$P_T^A(K^A) = \max_{E_M^A\geq 0 E_N^A\geq 0}\int_0^\infty e^{-\rho t}\left[\psi(K^A) - (1-\varphi_M-\tau)\frac{\mu_M}{2}(E_M^A)^2 - (1-\varphi_N-\sigma)\frac{\mu_N}{2}(E_N^A)^2\right]dt \tag{7}$$

Theorem 1. The equilibrium result under centralized decision-making is:

1. The optimal equilibrium strategy of creditor $M$ and debt enterprise $N$ is:

$$E_M^{A*} = \frac{\omega\alpha}{\mu_M(\rho+\gamma)(1-\varphi_M-\tau)}$$

$$E_N^{A*} = \frac{\omega\beta}{\mu_N(\rho+\gamma)(1-\varphi_N-\sigma)} \tag{8}$$

2. The optimal trajectory of synergistic effect of debt restructuring is:

$$K^{A*} = D^A - (D^A - K_0)e^{-\gamma t}$$

$$D^A = \frac{\omega\alpha^2}{\gamma\mu_M(\rho+\gamma)(1-\varphi_M-\tau)} + \frac{\omega\beta^2}{\gamma\mu_N(\rho+\gamma)(1-\varphi_N-\sigma)} \tag{9}$$

3. The optimal value of the total profit of the debt restructuring system is:

$$P_T^{A*}(E_M, E_N) = e^{-\rho t}V_T^A(K)$$

$$= e^{-\rho t}\left\{\frac{\omega K}{\rho + \gamma} + \frac{\psi_0}{\rho} + \frac{\omega^2}{\rho(\rho + \gamma)^2}\left[\frac{\alpha^2}{2\mu_M(1 - \varphi_M - \tau)} + \frac{\beta^2}{2\mu_N(1 - \varphi_N - \sigma)}\right]\right\}$$

$$V_T^A(K) = \frac{\omega K}{\rho + \gamma} + \frac{\psi_0}{\rho} + \frac{\omega^2}{\rho(\rho + \gamma)^2}\left[\frac{\alpha^2}{2\mu_M(1 - \varphi_M - \tau)} + \frac{\beta^2}{2\mu_N(1 - \varphi_N - \sigma)}\right] \quad (10)$$

Proof of Theorem 1. The dynamic random control method is used to solve, after time $t$, the optimal value function of long-term profit of creditors and debt enterprises is: $P_T^{A*}(E_M, E_N) = e^{-\rho t}V_T^A(K)$. $V_T^A(K)\prime$ fits the HJB equation for all $K \geq 0$.

$$\rho V_T(K) = \max_{E_M \geq 0 E_N \geq 0}\left[(\omega K + \psi_0) - (1 - \varphi_M - \tau)\frac{\mu_M}{2}(E_M)^2 - (1 - \varphi_N - \sigma)\frac{\mu_N}{2}(E_N)^2 + V_T^{A\prime}(K)(\alpha E_M(t) + \beta E_N(t) - \gamma K)\right] (11)$$

The optimal strategy for both parties is obtained from the solution of the first-order condition:

$$E_M^A = \frac{\alpha V_T^{A\prime}(K)}{\mu_M(1 - \varphi_M - \tau)}$$

$$E_N^A = \frac{\beta V_T^{A\prime}(K)}{\mu_N(1 - \varphi_N - \sigma)} \quad (12)$$

Substituting (12) into (11):

$$\rho V_T^A(K) = (\omega - \gamma V_T^{A\prime}(K))K + \psi_0 + \left[\frac{\alpha^2}{2\mu_M(1 - \varphi_M - \tau)} + \frac{\beta^2}{2\mu_N(1 - \varphi_N - \sigma)}\right](V_T^{A\prime}(K))^2 \quad (13)$$

Analysis of Formula (3) shows that the linear function of $K$ is the solution of the HJB equation. Assume $V_T^A(K) = aK + b$, where $a$ and $b$ are constants, we can get:

$$a = \frac{\omega}{\rho + \gamma}$$

$$b = \frac{\psi_0}{\rho} + \frac{\left[\frac{\alpha^2}{2\mu_M(1 - \varphi_M - \tau)} + \frac{\beta^2}{2\mu_N(1 - \varphi_N - \sigma)}\right]\left(\frac{\omega}{\rho + \gamma}\right)^2}{\rho}$$

$$= \frac{\psi_0}{\rho} + \frac{\omega^2}{\rho(\rho + \gamma)^2}\left[\frac{\alpha^2}{2\mu_M(1 - \varphi_M - \tau)} + \frac{\beta^2}{2\mu_N(1 - \varphi_N - \sigma)}\right] \quad (14)$$

Substitute Eq (14) into Eq (12) to obtain the equilibrium strategy of creditors and debtor enterprises under centralized decision-making, as shown in Eq (8); Then the optimal strategy (8) into Eq (2) to obtain the optimal trajectory of the synergistic effect of debt restructuring, as shown in Eq (9); Finally, substituting Eq (14) into $V_T^A(K) = aK + b$, and then the obtained $V_T^{A*}(K)$ into $P_T^{A*}(E_M, E_N) = e^{-\rho t}V_T^A(K)$, the total profit of the system can be obtained, as shown in Eq (10).

## Decentralized decision-making

Decentralized decision-making (represented by the superscript $B$) takes decision-making with the goal of maximizing the respective interests of the creditor and the debtor enterprise. At this time, the decision-making objectives are:

$$P_M^B(x) = \max_{E_M^B \geq 0} \int_0^\infty e^{-\rho t} \left[ \theta(\omega K^B + \psi_0) - \frac{\mu_M}{2}(E_M^B)^2 + \varphi_M \frac{\mu_M}{2}(E_M^B)^2 + \tau \frac{\mu_M}{2}(E_M^B)^2 \right] dt \quad (15)$$

$$P_N^B(x) = \max_{E_N^B \geq 0} \int_0^\infty e^{-\rho t} \left[ (1-\theta)(\omega K^B + \psi_0) - \frac{\mu_N}{2}(E_N^B)^2 + \varphi_N \frac{\mu_N}{2}(E_N^B)^2 + \sigma(E_N^B)^2 \right] dt \quad (16)$$

Theorem 2. The equilibrium result under decentralized decision-making is:

1. The optimal equilibrium strategy of creditor $M$ and debt enterprise $N$ is:

$$E_M^{B*} = \frac{\alpha \theta \omega}{\mu_M(\rho + \gamma)(1 - \varphi_M - \tau)}$$

$$E_N^{B*} = \frac{\omega \beta(1-\theta)}{\mu_N(\rho + \gamma)(1 - \varphi_N - \sigma)} \quad (17)$$

2. The optimal trajectory of synergistic effect of debt restructuring is:

$$K^{B*} = D^B - (D^B - K_0)e^{-\gamma t}$$

$$D^B = \frac{\alpha^2 \theta \omega}{\gamma \mu_M(\rho + \gamma)(1 - \varphi_M - \tau)} + \frac{\omega \beta^2 (1-\theta)}{\gamma \mu_N(\rho + \gamma)(1 - \varphi_N - \sigma)} \quad (18)$$

3. The optimal total profit value of creditor $M$ and, enterprise $N$ and the debt restructuring system is:

$$P_M^{B*} = e^{-\rho t} V_M^{B*}(K); P_N^{B*} = e^{-\rho t} V_N^{B*}(K); P_K^{B*} = e^{-\rho t}(V_M^{B*}(K) + V_N^{B*}(K))$$

$$V_M^{B*}(K) = \frac{\theta \omega K}{\rho + \gamma} + \frac{\theta \psi_0}{\rho} + \frac{\alpha^2 \theta^2 \omega^2}{2\mu_M \rho(\rho + \gamma)^2(1 - \varphi_M - \tau)} + \frac{\beta^2 \theta \omega^2(1-\theta)}{\rho \mu_N(\rho + \gamma)^2(1 - \varphi_N - \sigma)}$$

$$V_N^{B*}(K) = \frac{\omega(1-\theta)K}{\rho + \gamma} + \frac{\psi_{0-\theta\psi_0}}{\rho} + \frac{\beta^2(\omega - \theta\omega)^2}{2\rho \mu_N(1 - \varphi_N - \sigma)(\rho + \gamma)^2} + \frac{\alpha^2 \theta \omega^2(1-\theta)}{\rho \mu_{M(\rho+\gamma)^2}(1 - \varphi_M - \tau)} \quad (19)$$

## Stackelberg game decision after introducing cost-sharing contract

Assuming that creditor $M$ is the leader in the debt restructuring system and debt enterprise $N$ is the follower, under the decision-making situation after the introduction of the cost-sharing contract, in order to improve the enthusiasm of creditors and debtor enterprises to participate in debt restructuring, in addition to the government subsidy incentives and the favorable measures of asset management companies, creditors are also used to give certain incentives to

debtor enterprises, such as interest relief. That is, the creditors share the cost of debt restructuring efforts with the proportion of $\varepsilon$ for the debt enterprises, with $0 \leq \varepsilon \leq 1$. Under this situation, the decision-making objectives of creditors and debt enterprises are:

$$P_M(K) = \max_{E_M \geq 0} \int_0^\infty e^{-\rho t} \left[ \theta(\omega K + \psi_0) - \left(1 - \varphi_M - \tau \varepsilon\right) \frac{\mu_M}{2} E_M^2 - \varepsilon \frac{\mu_N}{2} E_N^2 \right] dt \tag{20}$$

$$P_N(K) = \max_{E_N \geq 0} \int_0^\infty e^{-\rho t} \left[ (1 - \theta)(\omega K + \psi_0) - \left(1 - \varphi_N - \sigma - \varepsilon\right) \frac{\mu_N}{2} E_M^2 \right] dt \tag{21}$$

Theorem 3. The equilibrium result of the decision after introducing the cost-sharing contract is:

1. The optimal effort cost ratio shared by debt enterprises for creditors, the optimal effort cost ratio shared by creditors for debt enterprises and the optimal effort level of creditors and debt enterprises are respectively:

$$\varepsilon^* = \frac{(3\theta - 1)(1 - \varphi_N - \sigma)}{\theta + 1}$$

$$E_M^* = \frac{\alpha \omega \theta}{\mu_M (\rho + \gamma)(1 - \varphi_M - \tau)}$$

$$E_N^* = \frac{\beta \omega (1 + \theta)}{2\mu_N (\rho + \gamma)(1 - \varphi_N - \sigma)} \tag{22}$$

2. The optimal trajectory of the synergistic effect of debt restructuring is:

$$K^* = D - (D - K_0)e^{-\gamma t}$$

$$D = \frac{\alpha^2 \omega \theta}{r \mu_M (\rho + \gamma)(1 - \varphi_M - \tau)} + \frac{\omega \beta^2 (1 + \theta)}{2\gamma \mu_N (\rho + \gamma)(1 - \varphi_N - \sigma)} \tag{23}$$

3. The optimal total profit value of creditor $M$ and enterprise $N$ and the debt restructuring system is:

$$P_M = e^{-\rho t} V_M(K); P_N = e^{-\rho t} V_N(K); P_K = e^{-\rho t} (V_M(K) + V_N(K))$$

$$V_M(K) = \frac{\theta \omega}{\rho + \gamma} K + \frac{\theta \psi_0}{\rho} + \frac{\alpha^2 \omega^2 (\theta + 1)^2}{8\rho \mu_N (1 - \varphi_M - \tau)(\rho + \gamma)^2} + \frac{\alpha^2 \omega^2 \theta^2}{2\rho \mu_M (1 - \varphi_N - \sigma)(\rho + \gamma)^2}$$

$$V_N(K) = \frac{\omega - \theta \omega}{\rho + \gamma} K + \frac{\psi_0 (1 - \theta)}{\rho} + \frac{\beta^2 \omega^2 (1 - \theta)(1 + \theta)}{4\rho \mu_M (1 - \varphi_N - \sigma)(\rho + \gamma)^2}$$
$$+ \frac{\beta^2 \omega^{2\theta} (1 - \theta)}{\rho \mu_N (1 - \varphi_M - \tau)(\rho + \gamma)^2} \tag{24}$$

Proof of Theorem 3. Using backward induction to analyze and solve the decision model of cost sharing. As a follower, the debt enterprise $N$ makes a decision on its optimal effort level after learning the debt restructuring effort level ($E_M$) and cost-sharing ratio ($\varepsilon$) of the creditor $M$. At this point, the game is transformed into the unilateral optimal control problem of debt enterprise $N$. Suppose that there is a continuous bounded differential function $V_N(K)$ which satisfies the HJB equation for all $K \geq 0$, as shown in Eq (25).

$$\rho V_N(K) = \max_{E_N \geq 0}\left[(1-\theta)(\psi_0 + \omega K) - \left(1 - \varphi_N - \sigma - \varepsilon\right)\frac{\mu_N}{2}E_N^2 + V_N'(K)(\alpha E_M + \beta E_N - \gamma K)\right] \quad (25)$$

The optimal strategy of debt enterprise is solved by first derivative:

$$E_N = \frac{\beta V_N'(K)}{\mu_N(1 - \varphi_N - \sigma - \varepsilon)} \quad (26)$$

Creditor $M$ considers that the debt enterprise $N$ will adopt its own strategy choice according to its own given strategy ($E_M, \varepsilon$), therefore, creditor $M$ will determine its own strategy ($E_M, \varepsilon$) according to its rational optimal strategy ($E_N$) of enterprise $N$. Similarly, assuming the existence of a continuous bounded differential function $V_M(K)$, satisfying the HJB equation for all $G \geq 0$.

$$\rho V_M(K) = \max_{E_M \geq 0}\left[\theta(\psi_0 + \omega K) - \left(1 - \varphi_M - \tau\right)\frac{\mu_M}{2}E_M^2 - \varepsilon\frac{\mu_N}{2}E_N^2 + V_M'(K)(\alpha E_M + \beta E_N - \gamma K)\right] \quad (27)$$

Substituting Eq (30) into Eq (31), we can get:

$$\rho V_M(K) = \max_{E_M \geq 0}\left[\theta(\psi_0 + \omega K) - (1 - \varphi_M - \tau)\frac{\mu_M}{2}E_M^2 - \frac{\varepsilon\beta^2 V_N'(K)}{2\mu_N(1 - \varphi_N - \sigma - \varepsilon)^2} + V_M'(K)\left(\alpha E_M + \frac{\beta^2 V_N'(K)}{\mu_M(1 - \varphi_N - \sigma - \varepsilon)} - \gamma K\right)\right] \quad (28)$$

The optimal strategy and cost-sharing ratio of the creditor are obtained by the first derivative:

$$E_M = \frac{\alpha V_M'(K)}{\mu_M(1 - \varphi_M - \tau)} \quad (29)$$

$$\varepsilon = \frac{(2V_M'(K) - V_N'(K))(1 - \varphi_N - \sigma)}{2V_M'(K) + V_N'(K)} \quad (30)$$

Substituting Eq (29) into Eq (27), we can get:

$$\rho V_M(K) = (\theta\omega - r V_M'(K)K) + \theta\psi_0 + \frac{\alpha^2 V_M'(K)^2}{2\mu_M(1 - \varphi_M - \tau)} - \varepsilon\frac{\mu_N}{2}E_N^2 \quad (31)$$

Substituting Eq (26) into Eq (25), we can get:

$$\rho V_N(K) = [\omega(1 - \theta) - r V_N'(K)]K + \psi_0 - \theta\psi_0 + \frac{\beta^2 V_N'(K)^2}{2\mu_N(1 - \varphi_N - \sigma - \varepsilon)} + V_N'(K)\alpha E_M \quad (32)$$

According to the analysis, the expression of the HJB equation solution is as follows: assume $V_M(K) = a_5 K + b_5$, $V_N(K) = a_6 K + b_6$, where $a_1$, $a_2$, $b_1$, $b_2$ are constants, and we can get:

$$a_5 = \frac{\theta\omega}{\rho + \gamma}$$

$$b_5 = \frac{\theta\psi_0}{\rho} + \frac{\alpha^2\omega^2(\theta+1)^2}{8\rho\mu_N(1-\varphi_M-\tau)(\rho+\gamma)^2} + \frac{\alpha^2\omega^2\theta^2}{2\rho\mu_M(1-\varphi_N-\sigma)(\rho+\gamma)^2}$$

$$a_6 = \frac{\omega - \theta\omega}{\rho + \gamma}$$

$$b_6 = \frac{\psi_0(1-\theta)}{\rho} + \frac{\beta^2\omega^2(1-\theta)(1+\theta)}{4\rho\mu_M(1-\varphi_N-\sigma)(\rho+\gamma)^2} + \frac{\beta^2\omega^{2\theta}(1-\theta)}{\rho\mu_N(1-\varphi_M-\tau)(\rho+\gamma)^2} \quad (33)$$

Substitute Eq (33) into Eqs (26) and (29) to obtain the equilibrium strategy of creditors and debt enterprises under decentralized decision-making, such as Eq (22); Then, the optimal strategy Eq (22) is substituted into Eq (2) to obtain the optimal trajectory of debt restructuring synergy effect, such as Eq (23); Finally, the Eq (A.4) is respectively substituted into $V_M(K) = a_5 K + b_5$ and $V_N(K) = a_6 K + b_6$, and then the obtained $V_M^B$ and $V_N^B$ are respectively substituted into $P_M = e^{-\rho t}V_M(K)$; $P_N = e^{-\rho t}V_N(K)$; $P_K = e^{-\rho t}(V_M(K) + V_N(K))$, the profit of both parties and the total profit of the system can be further obtained, as shown in Eq (24).

## Three kinds of decision results and discussion

As an effective method to solve dynamic problems, the differential game is often applied to environmental protection, pollution reduction, pricing and supply chain [19–23], and few scholars apply it to the research of debt restructuring. Based on the existing literature, this paper constructs a differential game model to study the optimal decisions of creditors and debt enterprises under different decision-making situations. In order to more intuitively compare the optimal equilibrium strategies of debt restructuring under the three situations of centralization, decentralization and the introduction of cost-sharing contract, Table 1 lists the parameters of equilibrium strategies.

By comparing the optimal debt restructuring effort level, debt restructuring synergy effect and total profit under the three situations, the following three propositions are obtained.

Proposition 1. compares the results under three kinds of decisions, and we know that: $E_N^{A*} > E_N^{B*} = E_N^*$; when $1/3 < \theta < 1$, $E_M^{A*} > E_M^* > E_M^{B*}$, $D^A > D > D^B$, $P_M > P_M^{B*}$, $P_N^{B*} > P_N$, $P_K^{A*} > P_K > P_K^{B*}$; when $0 < \theta \leq 1/3$, $E_M^{A*} > E_N^* > E_N^{B*}$, $D^A > D > D^B$, $P_M > P_M^{B*}$, $P_N^{B*} > P_N$, $P_K^{A*} > P_K > P_K^{B*}$. That is, under centralized decision-making, the debt restructuring effort level, the total profit of the system and the synergistic effect of debt restructuring are the

**Table 1. Comparison of differential game equilibrium strategies under different decision-making situations.**

| equilibrium strategy | centralized decision-making | decentralized decision-making | Steinberg decision-making |
|---|---|---|---|
| The effort cost of creditors | $\frac{\omega\alpha}{\mu_M(\rho+\gamma)(1-\varphi_M-\tau)}$ | $\frac{\alpha\theta\omega}{\mu_M(\rho+\gamma)(1-\varphi_M-\tau)}$ | $\frac{\alpha\omega\theta}{\mu_M(\rho+\gamma)(1-\varphi_M-\tau)}$ |
| The effort cost of debt enterprises | $\frac{\omega\beta}{\mu_N(\rho+\gamma)(1-\varphi_N-\sigma)}$ | $\frac{\omega\beta(1-\theta)}{\mu_N(\rho+\gamma)(1-\varphi_N-\sigma)}$ | $\frac{\beta\omega(1+\theta)}{2\mu_N(\rho+\gamma)(1-\varphi_N-\sigma)}$ |
| The effort cost shared by creditors for debt enterprises | — | — | $\frac{(3\theta-1)(1-\varphi_N-\sigma)}{\theta+1}$ |
| The effort cost shared by debt enterprises for creditors | — | — | — |

highest. This shows that centralized decision-making can enhance the enthusiasm of both parties to participate in debt restructuring and promote the total profit of the system. However, both parties should voluntarily implement centralized decision-making and meet constraints. That is, under centralized decision-making, the profits of both parties are higher than those of the other two decision-making modes.

Prove: $E_N^{A*} - E_N^{B*} = \frac{\omega\alpha(1-\theta)}{\mu_M(\rho+\gamma)(1-\varphi_M-\tau)} > 0$, therefore, the same can be proved, $E_N^{B*} = E_N^*$; $E_M^{A*} - E_M^{B*} = \frac{\theta\omega\beta}{\mu_N(\rho+\gamma)(1-\varphi_N-\sigma)} > 0$, the same can be proved that $E_M^* > E_M^{B*}$; $D^A - D^B = \frac{(1-\theta)}{\gamma\mu_M(\rho+\gamma)(1-\varphi_M-\tau)} + \frac{\theta\omega\beta^2}{\gamma\mu_N(\rho+\gamma)(1-\varphi_N-\sigma)} > 0$, therefore, the same can prove that $D>D^B$.

Proposition 2. Under the Stackelberg game, the cost-sharing ratio of creditor $M$ to debt enterprise $N$ is affected by the income ratio of creditors. The cost-sharing ratio of creditors to debt enterprises is affected by the income ratio of creditors. When the creditor's income ratio is $1/3<\theta<1$, $M$ shares part of the cost for $N$. At this time, the efforts level of $M$ and $N$ are higher than those of decentralized decision-making, which shows that the cost-sharing contract plays an incentive role. When the creditor's income ratio is $0<\theta\leq1/3$, $M$ does not share the cost of $N$, that is, it has no incentive effect. At this time, the efforts level of $M$ and $N$ are lower than those of decentralized decision-making.

Proposition 3. Under the three situations, the optimal effort level of both parties and the debt restructuring effort level of creditors and debt enterprises is in direct proportion to the influence coefficient of synergistic effect of debt restructuring, the impact of debt restructuring synergy on total income, the government subsidy rates for $M$ and $N$ and the asset management companies reduce the effort cost of enterprises and creditors. It is inversely proportional to the discount factor, the natural attenuation rate of synergistic effect and the effort cost coefficient of creditors and debt enterprises. Under decentralized decision-making, the optimal effort level of both parties is also in direct proportion to the distribution ratio of the total proceeds of debt restructuring between creditors and debt enterprises. In the Stackelberg game, the optimal effort level of the creditor is directly proportional to the income ratio of the creditor $M$. The optimal effort level of debt enterprises is directly proportional to the proportion of income obtained by debt enterprises $N$. In the three situations, the total profit of the system is proportional to the influence coefficient of debt restructuring synergy effect on total debt restructuring income, the influence coefficient of the level of efforts of both parties on the synergistic effect of debt restructuring and the government subsidy ratio; it is inversely proportional to the sum of the cost coefficients of both parties of the discount rate and the natural decay rate of the synergistic effect of debt restructuring. This is consistent with the research conclusion of Biglaiser et al. [24], that is, the government tends to use economic flexibility to balance the interests of all parties in debt restructuring.

Proposition 3 shows that when the government subsidy rates $\varphi_M$ and $\varphi_N$ are large and the proportion of preferential strength ratio given by asset management companies is large, the debt restructuring efforts of both parties and the total system profit will increase. Therefore, under centralized decision-making, the cost of debt restructuring can be reduced through government subsidies or incentives from asset management companies. This is consistent with Zabalza's [25] research conclusion, that is, the government needs to update the financial mechanism reasonably to solve the huge debt problem between the main bodies. Under decentralized decision-making, creditors and debt enterprises only pay attention to their respective incomes. The greater the proportion of their own income, the greater the initiative to participate in debt restructuring. Under the Steinberg game, creditors and debtors encourage each other and share costs. However, in order to make the creditor participate in the cost-sharing contract voluntarily, some constraints need to be satisfied, that is, when the condition $1/3<\theta<1$ is satisfied. At this time, the creditor's profit under the cost-sharing contract is greater

than that under the decentralized decision-making. The specific profit distribution scheme depends on their negotiating ability, and the cost-sharing agreement needs to be unanimously adopted and strictly implemented by both parties, so as to ensure the optimal profit value under the condition of realizing the cost-sharing contract.

Inference 1. Combining Proposition 1, Proposition 2, Proposition 3, and Table 1 shows that the optimal feedback equilibrium strategies under the three decisions are all time-independent parameters. That is, the optimal feedback equilibrium strategy does not change with time, which is consistent with the actual situation. And from a dynamic point of view, it is verified that under the decision-making situation after the introduction of the cost-sharing contract, the efforts cost of both parties and the total profit of the debt restructuring system have been improved, and reaching the level of centralized decision-making, and achieving long-term win-win cooperation among all parties in debt restructuring, which is consistent with Yulian-to's [26] research conclusion, that is, adopting correct policy guidance to maximize the profits of all parties in debt restructuring. When the market is more mature, the operation mode of the debt restructuring system under this decision tends to be stable and the strategy after introducing the cost-sharing contract is feasible in the actual debt restructuring process. The model has good management practical significance, which provides an effective theoretical basis for the successful decision of debt restructuring.

Inference 2. In centralized decision-making, the synergistic effect and total profit of debt restructuring of creditors and debt enterprises are higher than those in decentralized decision-making. The Pareto improvement effect of cost-sharing contract on the total profits of both parties is affected by effort cost coefficient of creditor $\mu_M$ and debt enterprise $\mu_N$ and the influence coefficients $\alpha$ and $\beta$ of the debt restructuring efforts of creditors and debt enterprises on the synergistic effect of debt restructuring. The influence relationship of some parameters will be analyzed in an example.

## Example analysis

In order to further study the decision-making results in the above three situations, we will examine the influence of parameters on the results in this section. In this paper, Matlab2018 software is used for simulation to verify the validity of the model. Considering the complexity of the model, the relevant parameters in the example are assigned by reference [27–29], as follows: $\theta = 0.5$, $\mu_M = 20$, $\mu_N = 15$, $\alpha = 0.5$, $\beta = 0.3$,
$\gamma = 0.2$, $\omega = 30$, $\rho = 0.8$, $\psi_0 = 20$, $K_0 = 0$, $\tau = 0.1$, $\sigma = 0.2$, $\varphi_M = 0.2$, $\varphi_N = 0.1$, $t = 1$.

### Discussion on game equilibrium results

In this paper, the above parameters are substituted into the relevant propositions, and the game results of centralized decision-making, decentralized decision-making, and Stackelberg game decision after introducing the cost-sharing contract are obtained, as shown in Table 2.

Table 2 shows that: (1) compared with centralized decision-making situation, the effort level of creditors and debt enterprises under decentralized decision-making has decreased by 50% and 50.5%, respectively; the synergistic effect of debt restructuring decreased by 50.6%; the total profit from debt restructuring decreased by 6.1%. This shows that creditors and debtors only focus on their own interests under decentralized decision-making, which is not conducive to the process of debt restructuring; (2) After the introduction of the cost-sharing contract, the effort level of creditors and debt enterprises, the synergistic effect of debt restructuring and their respective profits have been improved, debt restructuring total profit compared to decentralized decision-making increased by 4.2%, almost reaching the level of

**Table 2. Differential game equilibrium results under different decisions.**

| | centralized decision-making | decentralized decision-making | Introducing cost sharing contract decision-making |
|---|---|---|---|
| **The effort cost of creditors** | 1.07 | 0.53 | 0.53 |
| **The effort cost of debt enterprises** | 0.85 | 0.42 | 0.64 |
| **The synergistic effect of debt restructuring** | 0.79 | 0.39 | 0.45 |
| **cost assignment ratio** | — | — | 0.23 |
| **Profit of debt Creditors** | — | 17.42 | 18.92 |
| **Profit of debt enterprises** | — | 18.72 | 18.72 |
| **Total profit of debt restructuring system** | 38.5 | 36.14 | 37.64 |

centralized decision-making, which indicates that the introduction of cost-sharing contract has effectively promoted the process of the debt restructuring system.

## Changes over time in three game situations

Figs 1 and 2 are the curves of the synergistic effect of debt restructuring and the total profit of the system changing with time under three game situations.

As shown in Fig 1, the optimal trajectory of the debt restructuring synergy effect in three situations is the maximum in centralized decision-making and the minimum in decentralized

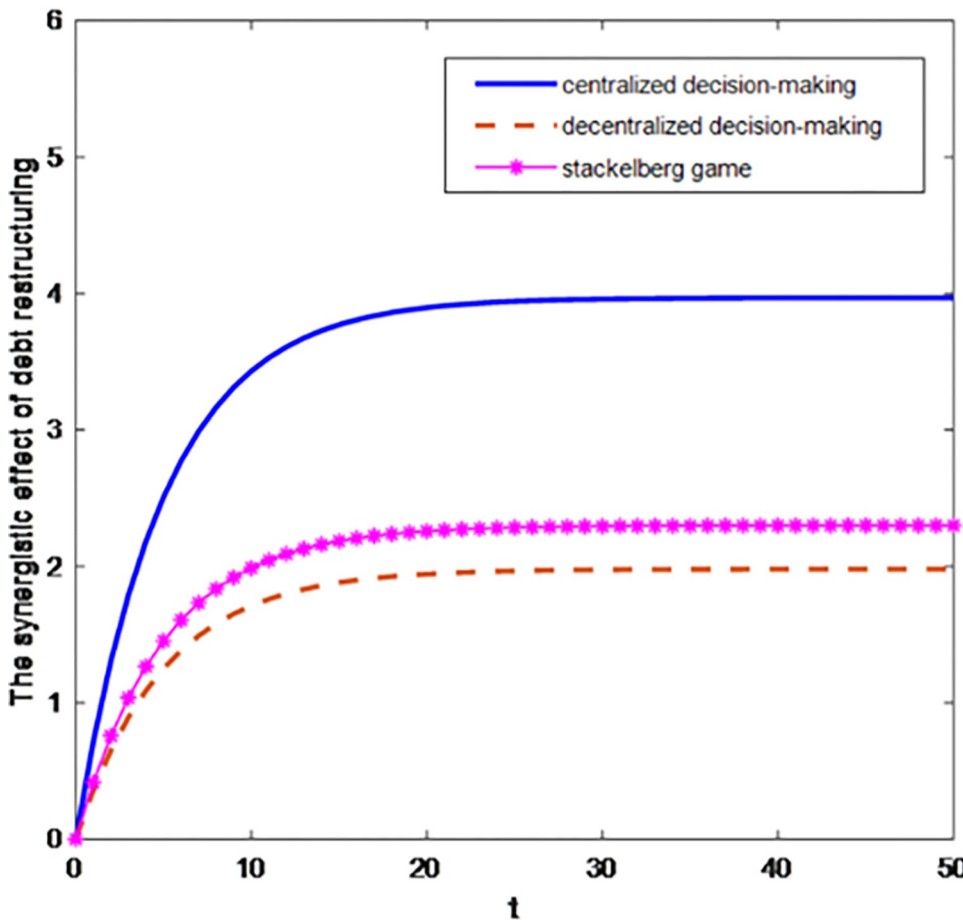

**Fig 1. The synergistic effect of debt restructuring under three game situations.**

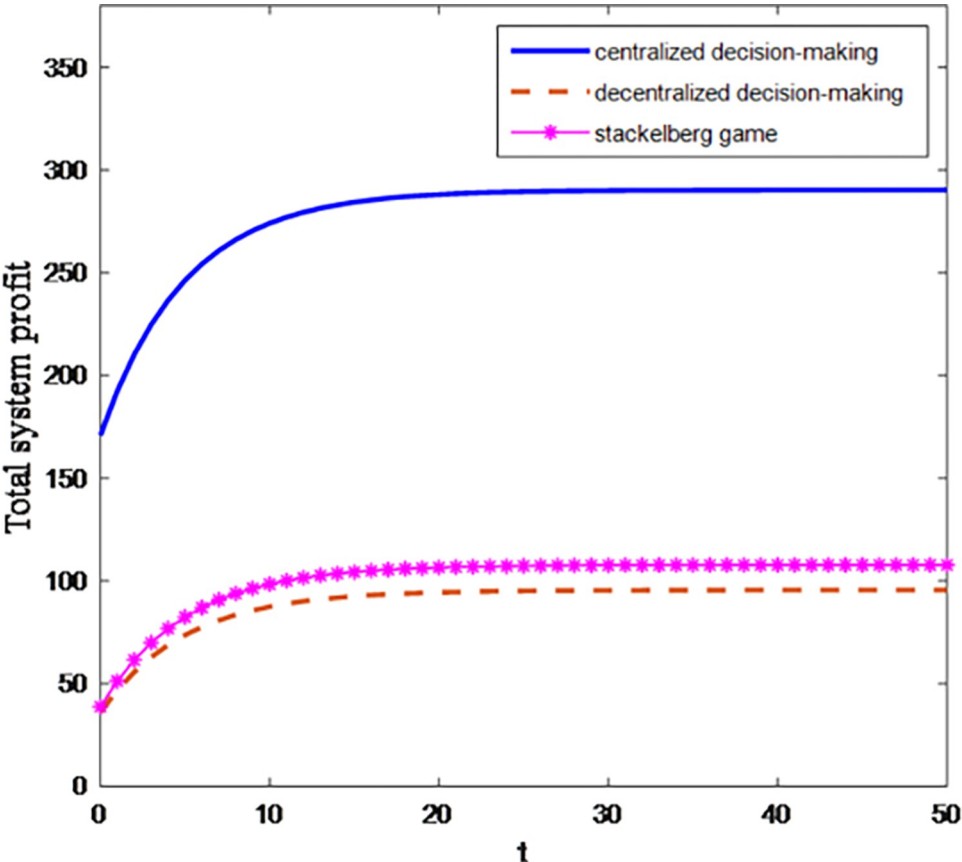

**Fig 2. Total profit of the system under three game situations.**

decision-making. The introduction of cost-sharing contract can achieve Pareto improvement of debt restructuring synergy. The synergistic effect of debt restructuring in centralized decision-making is much more significant than that in decentralized decision-making, which verifies the results of theoretical derivation. Further analysis shows that the synergy effect of debt restructuring is growing rapidly in centralized decision-making. The synergy effect of debt restructuring increases slowly in decentralized decision-making. This shows that centralized decision-making is better than decentralized decision-making, which can provide the reference for all parties to participate successfully in debt restructuring.

Fig 2 shows that with the time, the total profit of debt restructuring system under centralized decision-making is always higher than that under decentralized decision-making, while Stackelberg game under cost-sharing contract can improve the equilibrium result of decentralized decision-making compared with decentralized decision-making, and the total profit of debt restructuring system under cost-sharing contract is improved. This shows that after the introduction of cost-sharing contract, the effort level of debt restructuring is more easily transformed into the total profit of debt restructuring system. The more obvious the effect of creditors and debt enterprises participating in debt restructuring, the total profit of debt restructuring system is higher than that of decentralized decision-making, and the effort of debt restructuring has a greater impact on the total profit of debt restructuring system.

Therefore, the use of cost sharing contract can achieve Pareto improvement of creditors and debt enterprises, and the introduction of cost sharing contract has more obvious advantages than decentralized decision-making in promoting debt restructuring process. The

improvement effect of cost sharing contract on the synergistic effect of debt restructuring is better than that on the total profit of debt restructuring system. The change of slope before and after the introduction of cost sharing contract can also reflect this point.

### Sensitivity analysis of related parameters

To deeply study the impact of changes in relevant parameters of enterprise debt restructuring on the synergy effect of debt restructuring and the total profit of the system under centralized decision-making, decentralized decision-making and the Stackelberg game under cost-sharing contract, this paper selects some parameters for sensitivity analysis.

As shown in Figs 3 and 4, under the centralized decision-making, with the increase of the influence degree $\omega$ of debt restructuring synergy on the total profit, the debt restructuring synergy and the total profit of debt restructuring both increase. The optimal trajectory and the total profit of the synergistic effect of debt restructuring increase with time and tend to be stable, that is, the increase rate of the synergy effect of debt restructuring and the total profit of debt restructuring is getting smaller and smaller. This shows that the synergistic effect of debt restructuring value and the total profit value increase within a certain range, and at any same time, the corresponding slope of the optimal trajectory of debt restructuring synergy and the total profit increase with the increase of the sensitivity coefficient of total profit. At this time, creditors and debt enterprises have the highest efficiency in debt restructuring.

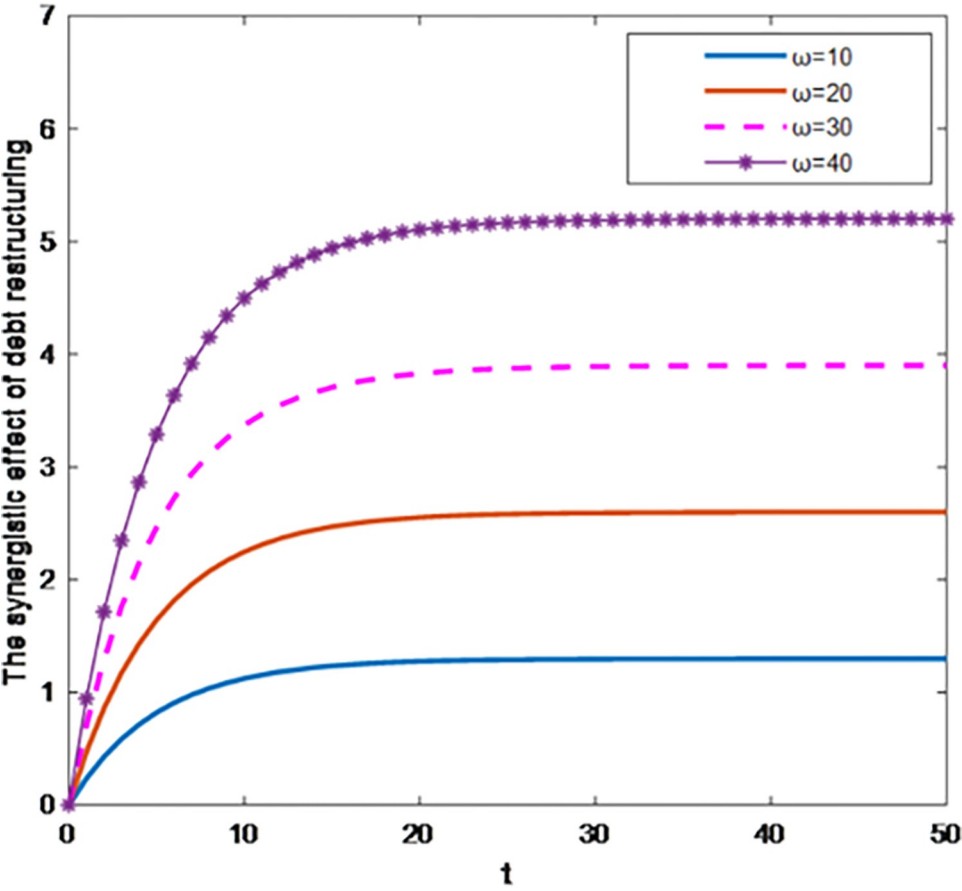

**Fig 3. Effect of parameter $\omega$ on the synergistic effect of debt restructuring.**

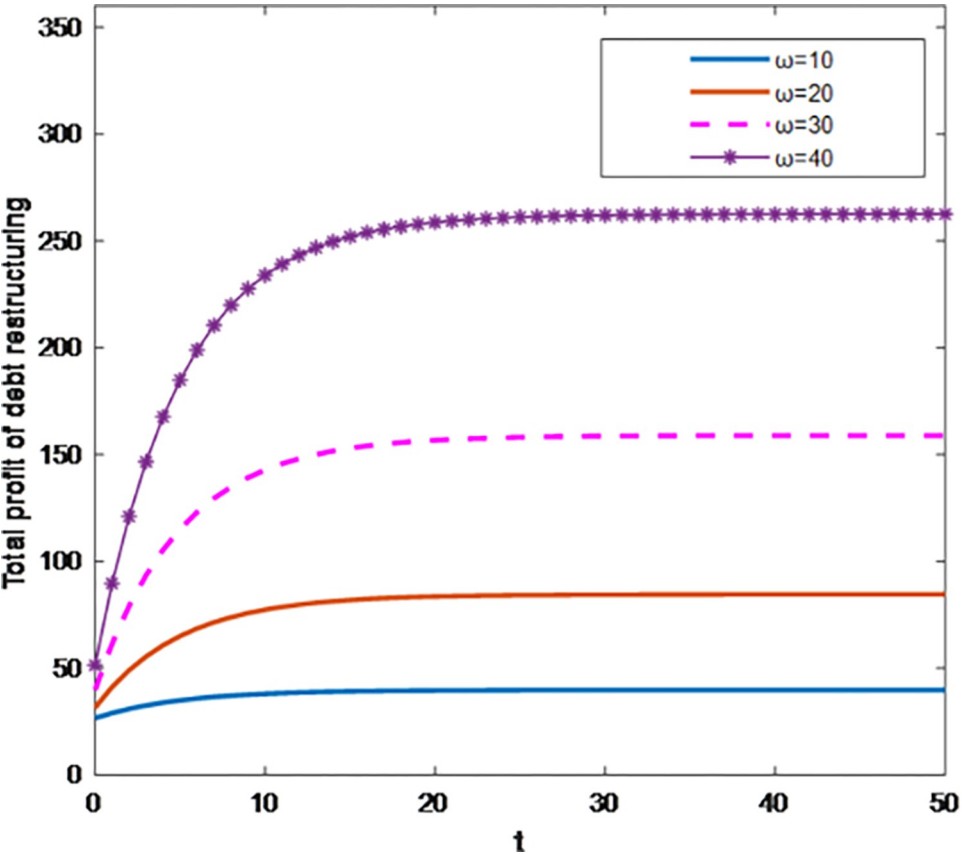

**Fig 4. Effect of parameter *ω* on total profit of debt restructuring.**

As shown in Fig 5, under centralized decision-making, the synergistic effect of debt restructuring has a natural decay rate. The greater the natural decay rate r, the smaller the value of the optimal trajectory of the synergistic effect of debt restructuring at the same time point, and it gradually stabilizes on the horizontal line. At this time, the effect of creditors and debt enterprises participating in debt restructuring is less obvious, which shows that when the natural decay rate is relatively large, it will hinder the process of debt restructuring.

As shown in Fig 6, in the three game situations, the synergistic effect of debt restructuring increases with the increase of parameter *α*, while the synergistic effect of debt restructuring under centralized decision-making is higher than that under decentralized decision-making, and the Stackelberg game under cost-sharing contract can improve the equilibrium result of decentralized decision-making, which shows that the greater the level of debt restructuring efforts, the more it can promote the debt restructuring process.

As shown in Fig 7, the synergistic effect of debt restructuring increases with the increase of the effort cost coefficient of creditors and debtors, but due to the different settings of other parameters of different enterprises, the effort cost coefficient of creditors and debtors has different degrees of influence on the synergistic effect of debt restructuring.

As shown in Fig 8, the total profit of debt restructuring system will increase with the increase of the effort cost coefficient of creditors and debtors, but the effort cost coefficient of creditors and debt enterprises has different effects on the total profit of debt restructuring, respectively. Due to the different settings of other parameters of different enterprises, the effort

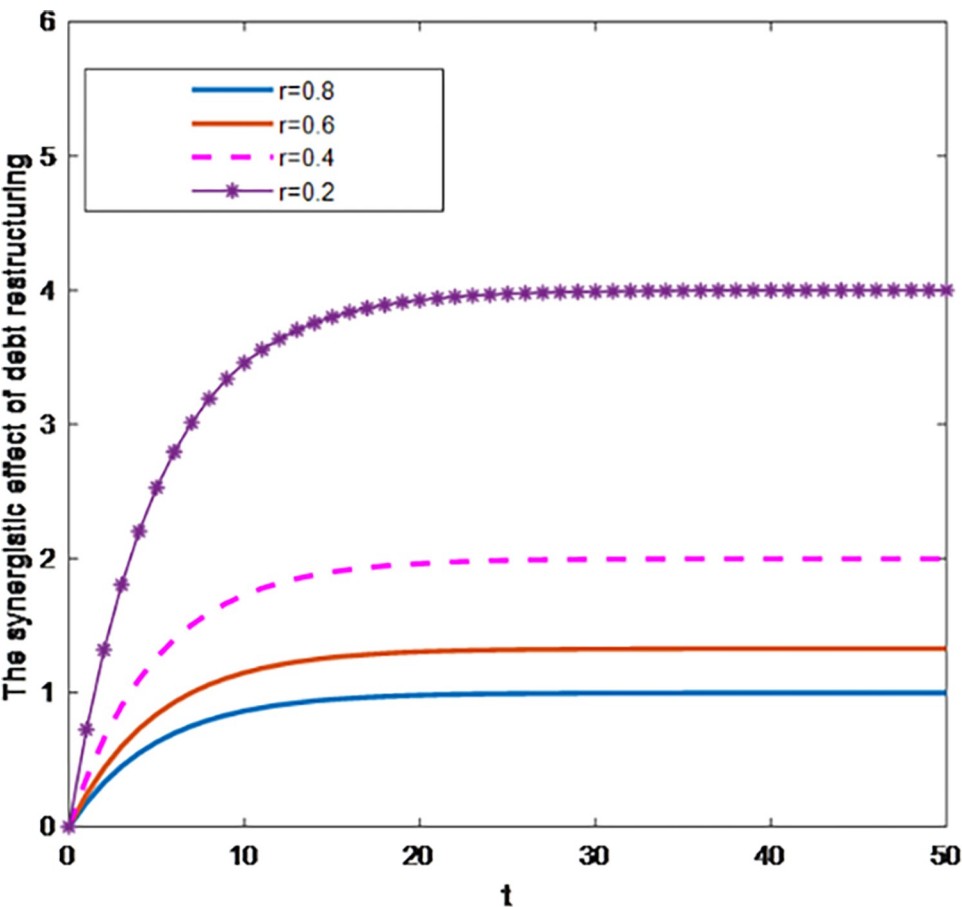

**Fig 5. Influence of parameter r on the synergistic effect of debt restructuring.**

cost coefficients of creditors and debtors have different degrees of influence on the synergistic effect of debt restructuring.

## Government subsidy policy analysis

Figs 9 and 10 show the influence of government subsidy rate on the synergistic effect of debt restructuring and the total profit of the system under centralized decision-making.

This paper analyzes the influence of government support policies on the synergistic effect of debt restructuring between creditors and debt enterprises and the total profit of debt restructuring system, and makes sensitivity analysis on $\varphi_M$ and $\varphi_N$ under centralized decision, as shown in Figs 9 and 10.

As shown in Fig 9, the synergistic effect of debt restructuring increases with the increase of government subsidies to creditors and debt enterprises, which shows that government support can encourage all subjects of debt restructuring to actively participate in debt restructuring. Therefore, the government can improve the synergistic effect of debt restructuring by increasing the subsidies to creditors and debtors, thus promoting the process of debt restructuring. However, the subsidies of other parameters will have different degrees of influence on the synergistic effect of debt restructuring.

As shown in Fig 10, with the increase of the government's support for debt restructuring, the total profit of the debt restructuring system also increases with the increase of the

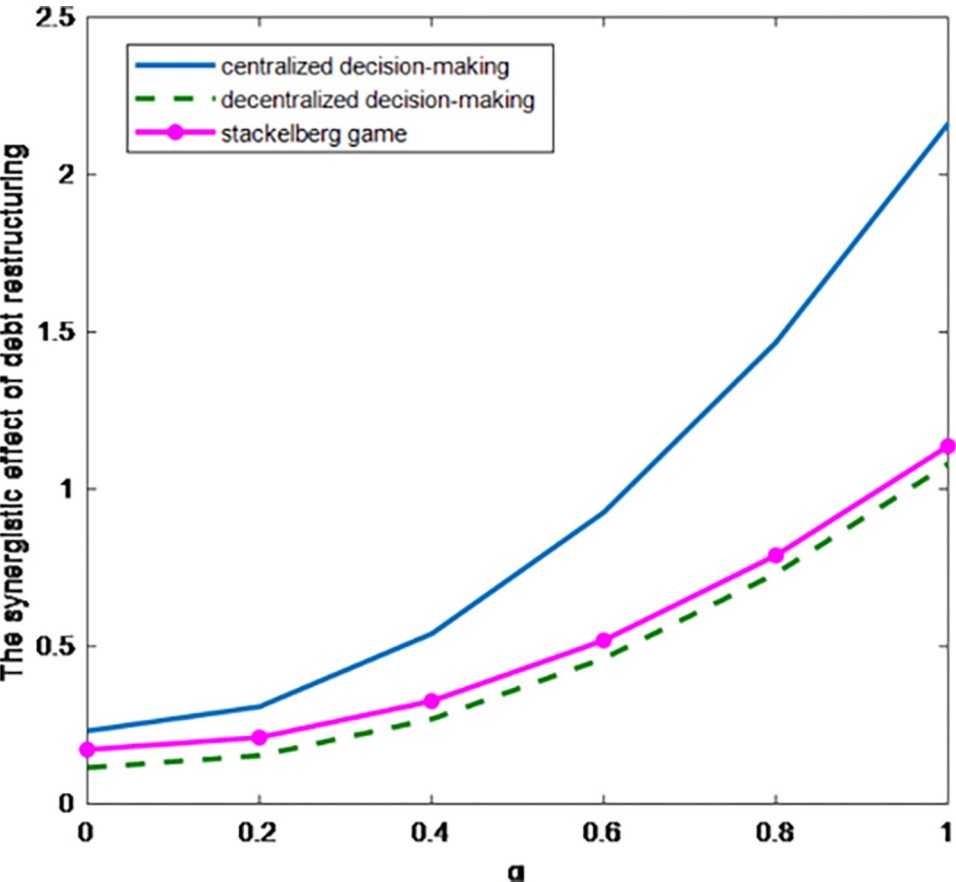

**Fig 6. Influence of parameter α on the synergistic effect of debt restructuring.**

government's subsidy rate. As the leader in the debt restructuring system, creditors can benefit from government subsidies to debt enterprises while giving them preferential treatment. Therefore, government subsidies can help debt restructuring better coordinate development. This shows that the government can increase the total profit of the system by increasing the degree of subsidies to creditors and debtors, but the subsidy rates to creditors and debtors will have different degrees of influence on the synergistic effect of debt restructuring.

## Conclusion

This paper uses differential game theory to study the problem of debt restructuring composed of creditors and debt enterprises, and discusses the influence of government subsidies and the participation of asset management companies on the debt restructuring system. Considering that the synergistic effect of debt restructuring has naturally declined over time, this paper constructs a differential game model of debt restructuring in three situations: centralized decision-making, decentralized decision-making and decision-making after introducing cost-sharing contract. Through the simulation method, the sensitivity analysis of related parameters is carried out, and the optimal decision results in three situations are compared and analyzed, and some conclusions are obtained as follows:

1. Compared with decentralized decision-making, the optimal effort level of creditors and debt enterprises, the optimal synergy effect trajectory and its stable value of debt

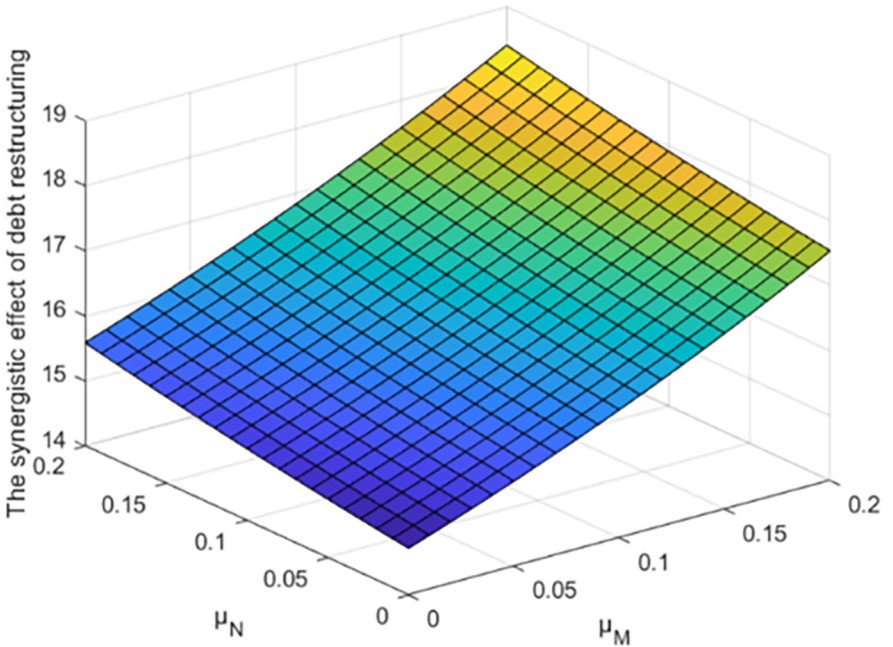

**Fig 7. Effect of parameter $\mu_M$ and $\mu_N$ on the synergistic effect of debt restructuring.**

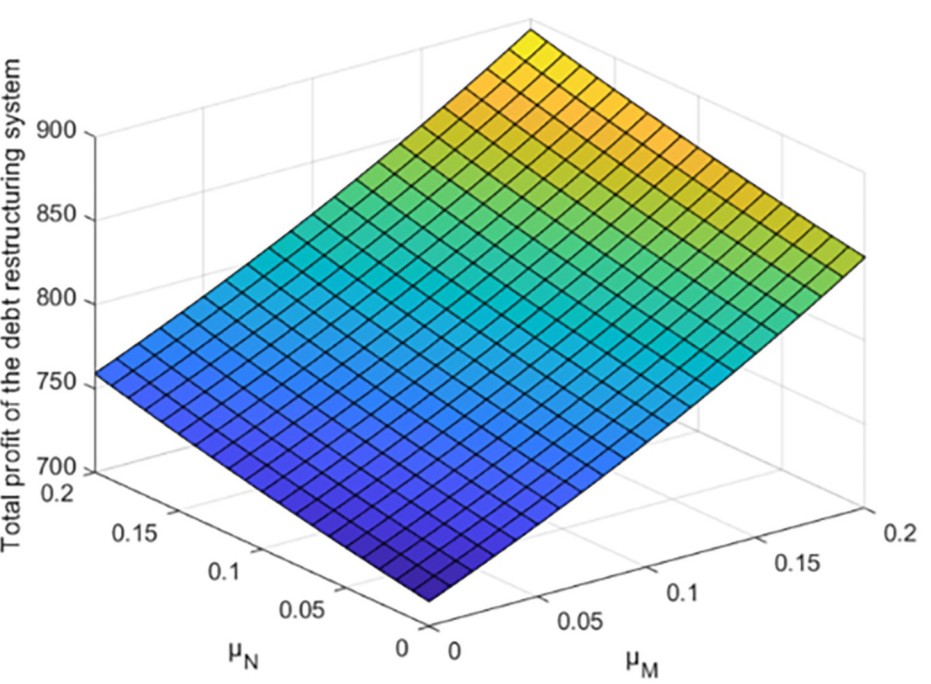

**Fig 8. Effect of parameter $\mu_M$ and $\mu_N$ on total profit of debt restructuring.**

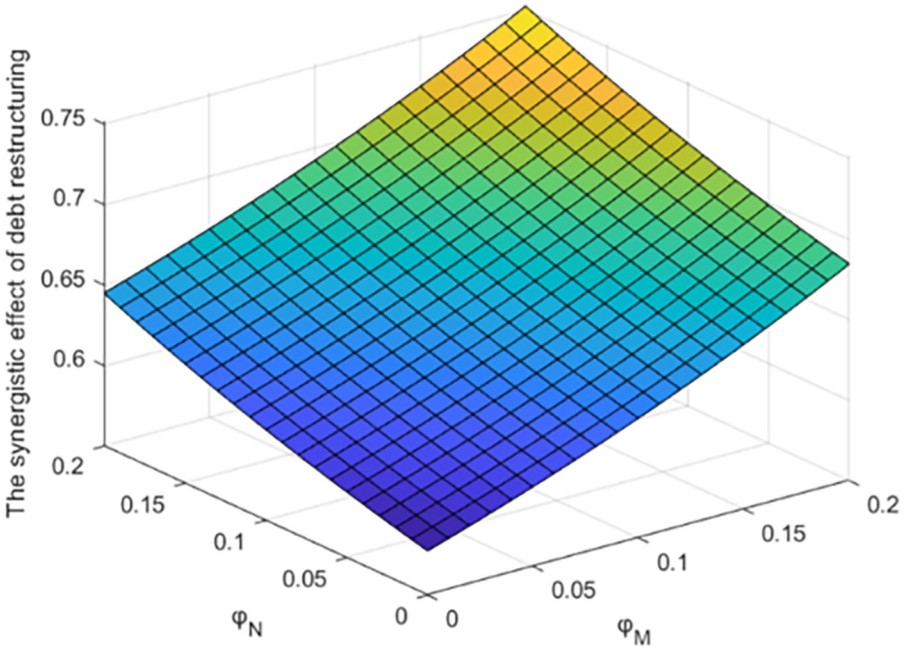

**Fig 9. Effect of parameter $\varphi_M$ and $\varphi_N$ on the synergistic effect of debt restructuring.**

restructuring system and the long-term profit of debt restructuring system under centralized decision-making are improved. With the increase of the synergistic effect of debt restructuring, the interests of all participants in debt restructuring will also increase accordingly, which will stimulate the enthusiasm of both parties.

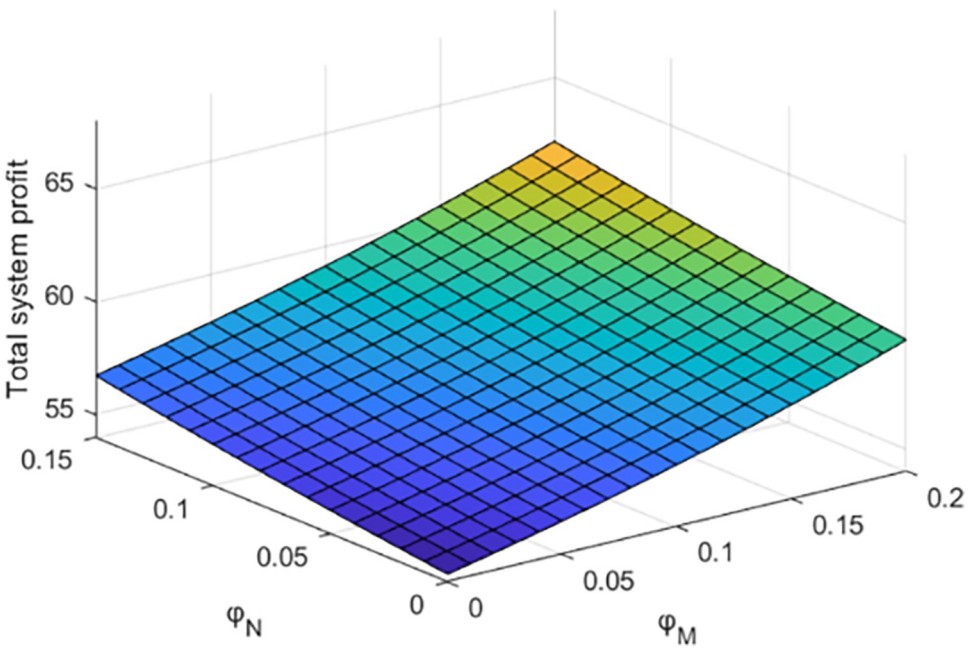

**Fig 10. Effect of parameter $\varphi_N$ and $\varphi_N$ on total profit of debt restructuring.**

2. With the introduction of the cost-sharing contract, the optimal income of both parties and the overall income of debt restructuring are better than decentralized decision-making, which shows that "cost-sharing", as an effective adjustment mechanism, can promote the synergy of debt restructuring and increase the income of both participants, thus realizing Pareto optimality of debt restructuring system.

3. Under the Stackelberg game, when the proportion of creditors' income is $1/3 < \theta < 1$, the efforts of creditors and debt enterprises are higher than those of decentralized decision-making, which shows that the cost-sharing contract has played an incentive role. When the proportion of the creditor's income is $0 < \theta \leq 1/3$ the creditor does not share the cost of the debt enterprise at this time, and there is no incentive effect at this time.

4. The introduction of cost-sharing contracts has improved the efforts of creditors and debt enterprises, the synergy of debt restructuring, and the profits of debt restructuring, and the total profit of debt restructuring almost reached the level of centralized decision-making situation. Therefore, the introduction of the cost-sharing contract effectively realizes the coordination of the debt restructuring system, and not only maximizes the profit of the debt restructuring system, but also improves the debt restructuring environment and promotes the debt restructuring process.

5. With the increase of the cost coefficient of creditors and debt enterprises and the natural decay rate of debt restructuring, the synergistic effect of debt restructuring shows a downward trend. This shows that the increase in the cost invested by creditors and debt enterprises, it will hinder the enthusiasm of creditors and debt enterprises to participate in debt restructuring, reducing the efficiency of debt restructuring, which is consistent with the actual situation. At this time, the enthusiasm of creditors and debt enterprises can be improved by introducing government subsidies and the participation of asset management companies, which provides a scientific theoretical reference for the government and asset management companies to participate successfully in debt restructuring.

6. The government subsidy rate and the preferential strength given by asset management companies will affect the process of debt restructuring. When the government support increases, the level of debt restructuring efforts of both sides and the total profits of the system will increase. Therefore, under centralized decision-making, the cost of debt restructuring can be reduced through government subsidies or incentives from asset management companies. The government should strengthen tax reduction for endangered enterprises and lower the threshold of debt restructuring, thus improving the success rate of debt restructuring. Under the Stackelberg game, creditors and debt enterprises encourage each other and share costs with each other The cost-sharing agreement needs to be unanimously approved and strictly implemented by both parties to ensure the optimal profit value under the cost-sharing contract. The research in this paper has a positive guiding significance for the decision-making of the main body of debt restructuring and the policy making of the government.

## Supporting information

**S1 Data. Availability statement.**
(DOCX)

**S1 File.**
(DOCX)

**S2 File.**
(DOCX)

## Author Contributions

**Conceptualization:** Danyu Zhao, Li Song.

**Data curation:** Danyu Zhao.

**Formal analysis:** Liangliang Han.

**Funding acquisition:** Liangliang Han.

**Investigation:** Liangliang Han.

**Methodology:** Danyu Zhao.

**Project administration:** Liangliang Han.

**Resources:** Li Song.

**Software:** Danyu Zhao, Li Song.

**Writing – original draft:** Danyu Zhao, Li Song.

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
