## [Decision Letter · Decision Letter 0]

8 Jan 2023

PONE-D-22-29886

Research on Differential Game Strategy of Debt Restructuring Supported by Government

PLOS ONE

Dear Dr. Song,

Thank you for submitting your manuscript to PLOS ONE. After careful consideration, we feel that it has merit but does not fully meet PLOS ONE’s publication criteria as it currently stands. Therefore, we invite you to submit a revised version of the manuscript that addresses the points raised during the review process.

We look forward to receiving your revised manuscript.

Kind regards,

Muhammad Fareed, Ph.D

Academic Editor

PLOS ONE

Journal Requirements:

   "This research was funded by National Social Science Foundation of China, grant number 21BGL110."

Additional Editor Comments:

Dear Author,

Please make the amendments as per the followings:

1. Abstract is poorly written and need extensive improvement

2. The very first Paragraph is without any reference

3. The introduction part is very weak and seems like literature review. I suggest improving it according to the title of the study

4. Motivation and objectives of the study need more focus

5. A summary of previous related studies is to be given in one Table form at the end of the Introduction part. Take help of these close related studies:

- External debt - blessing or curse: empirical evidence from Pakistan. International Journal of Economics and Financial Issues.

- External debt and public investment: a case study of Pakistan. Journal of Managerial Sciences,

-The role of external debt in economic growth of Indonesia – A Blessing or Burden?

- Does military expenditure increase external debt? Evidence from Asia. Defence and Peace Economics

-Estimating the Optimum Level of Debt-Threshold: Empirical Evidence from Pakistan. Pakistan Social Sciences Review.

6. Source at the end of Table needs to be given where needed

7. Shed more lights on the discussion on results

8. Focus more on the policy recommendations.

9. Highlight main contributions of the study

10. The discussion of the findings is missing. Please contextualize your findings with the literature.

11. The motivation and rationale of the study needs further attention by the authors.

12. The manuscript is very less reader friendly. The theoretical discussion can be moved to the annexure.

13. Please explain how this work stands out from the literature in the said field. How it can be assist in policy design.

Reviewers' comments:

Reviewer's Responses to Questions

**Comments to the Author**

1. Is the manuscript technically sound, and do the data support the conclusions?

Reviewer #1: Yes

Reviewer #2: Yes

2. Has the statistical analysis been performed appropriately and rigorously? 

Reviewer #1: Yes

Reviewer #2: Yes

3. Have the authors made all data underlying the findings in their manuscript fully available?

Reviewer #1: Yes

Reviewer #2: Yes

4. Is the manuscript presented in an intelligible fashion and written in standard English?

Reviewer #1: Yes

Reviewer #2: Yes

5. Review Comments to the Author

Reviewer #1: The Editor,

PLOS ONE

I have reviewed overall manuscript on “Research on Differential Game Strategy of Debt Restructuring Supported by Government”. Following are my comments on the manuscript:

1. Abstract is poorly written and need extensive improvement

2. The very first Paragraph is without any reference

3. The introduction part is very weak and seems like literature review. I suggest improving it according to the title of the study

4. Motivation and objectives of the study need more focus

5. A summary of previous related studies is to be given in one Table form at the end of the Introduction part. Take help of these close related studies:

- External debt - blessing or curse: empirical evidence from Pakistan. International Journal of Economics and Financial Issues.

- External debt and public investment: a case study of Pakistan. Journal of Managerial Sciences,

-The role of external debt in economic growth of Indonesia – A Blessing or Burden?

- Does military expenditure increase external debt? Evidence from Asia. Defence and Peace Economics

-Estimating the Optimum Level of Debt-Threshold: Empirical Evidence from Pakistan. Pakistan Social Sciences Review.

6. Source at the end of Table needs to be given where needed

7. Shed more lights on the discussion on results

8. Focus more on the policy recommendations.

9. Highlight main contributions of the study

Reviewer #2: I am pleased to review the manuscript titled "Research on Differential Game Strategy of Debt Restructuring Supported by Government" in which authors tried to investigate the debt restructuring equilibrium strategy through differential game method. The findings of the study are interesting particularly related to the synergy effect and total profit of debt restructuring under centralized decision-making, and the synergy effect and total profit of debt restructuring after the introduction of cost-sharing. The following comments will help to enhance the quality of the manuscript:

1. The discussion of the findings is missing. Please contextualize your findings with the literature.

2. The motivation and rationale of the study needs further attention by the authors.

3. The manuscript is very less reader friendly. The theoretical discussion can be moved to the annexure.

4. Please explain how this work stands out from the literature in the said field. How it can be assist in policy design.

6. PLOS authors have the option to publish the peer review history of their article (what does this mean?). If published, this will include your full peer review and any attached files.

Reviewer #1: No

Reviewer #2: No

---

## [Author Response · Author response to Decision Letter 0]

28 Feb 2023

Thanks for your letter and for the reviewers’comments concerning our manuscript. The comments are all valuable and very helpful for improving the quality of our paper. We agree with the comments and have made amendments accordingly. The changes made in the manuscript are highlighted in blue in the paper. Please refer to the file labeled 'Response to Reviewers' for the specific information of the paper modification.

---

## [Decision Letter · Decision Letter 1]

22 Mar 2023

Research on Differential Game Strategy of Debt Restructuring Supported by Government

PONE-D-22-29886R1

Dear Dr. Li Song,

We’re pleased to inform you that your manuscript has been judged scientifically suitable for publication and will be formally accepted for publication once it meets all outstanding technical requirements.

Kind regards,

Muhammad Fareed, Ph.D

Academic Editor

PLOS ONE

Additional Editor Comments (optional):

Dear Author/s,

Thank you for making the corrections as per reviewers' comments.

We are delighted to inform you that your article is accepted.

Thank you.

Reviewers' comments:

Reviewer's Responses to Questions

**Comments to the Author**

1. If the authors have adequately addressed your comments raised in a previous round of review and you feel that this manuscript is now acceptable for publication, you may indicate that here to bypass the “Comments to the Author” section, enter your conflict of interest statement in the “Confidential to Editor” section, and submit your "Accept" recommendation.

Reviewer #1: All comments have been addressed

Reviewer #2: All comments have been addressed

2. Is the manuscript technically sound, and do the data support the conclusions?

Reviewer #1: Yes

Reviewer #2: Yes

3. Has the statistical analysis been performed appropriately and rigorously? 

Reviewer #1: Yes

Reviewer #2: Yes

4. Have the authors made all data underlying the findings in their manuscript fully available?

Reviewer #1: Yes

Reviewer #2: Yes

5. Is the manuscript presented in an intelligible fashion and written in standard English?

Reviewer #1: Yes

Reviewer #2: Yes

6. Review Comments to the Author

Reviewer #1: I have reviewed the revised manuscript and found that the authors have been successfully addressed al suggested.

If possible, edit language where needed.

Reviewer #2: The authors have addressed all the comments and recommendations have been considered well in the revised version of the manuscript.

7. PLOS authors have the option to publish the peer review history of their article (what does this mean?). If published, this will include your full peer review and any attached files.

Reviewer #1: No

Reviewer #2: No

---

## [Editor Report · Acceptance letter]

28 Mar 2023

PONE-D-22-29886R1 

Research on Differential Game Strategy of Debt Restructuring Supported by Government 

Dear Dr. Song:

I'm pleased to inform you that your manuscript has been deemed suitable for publication in PLOS ONE. Congratulations! Your manuscript is now with our production department. 

Kind regards, 

on behalf of

Dr. Muhammad Fareed 

Academic Editor

PLOS ONE